# A class-mismatched TCR bypasses MHC restriction via an unorthodox but fully functional binding geometry

Nishant K. Singh[1,3], Jesus A. Alonso[1,4], Jason R. Devlin[1,5], Grant L. J. Keller [1,6], George I. Gray[1], Adarsh K. Chiranjivi[1], Sara G. Foote[1], Lauren M. Landau [1,7], Alyssa G. Arbuiso[1], Laura I. Weiss [1], Aaron M. Rosenberg[1], Lance M. Hellman[1,8], Michael I. Nishimura[2] & Brian M. Baker [1] ✉

MHC restriction, which describes the binding of TCRs from CD4+ T cells to class II MHC proteins and TCRs from CD8+ T cells to class I MHC proteins, is a hallmark of immunology. Seemingly rare TCRs that break this paradigm exist, but mechanistic insight into their behavior is lacking. TIL1383I is a prototypical class-mismatched TCR, cloned from a CD4+ T cell but recognizing the tyrosinase tumor antigen presented by the class I MHC HLA-A2 in a fully functional manner. Here we find that TIL1383I binds this class I target with a highly atypical geometry. Despite unorthodox binding, TCR signaling, antigen specificity, and the ability to use CD8 are maintained. Structurally, a key feature of TIL1383I is an exceptionally long CDR3β loop that mediates functions that are traditionally performed separately by hypervariable and germline loops in canonical TCR structures. Our findings thus expand the range of known TCR binding geometries compatible with normal function and specificity, provide insight into the determinants of MHC restriction, and may help guide TCR selection and engineering for immunotherapy.

αβ T cells orchestrate cellular immunity by using their heterodimeric T-cell receptor, or TCR, to recognize foreign and self-antigens. Although they resemble the antigen-binding fragments of antibodies, TCRs differ in that they are restricted towards antigens bound and presented by major histocompatibility complex (MHC) proteins. MHC class restriction is a hallmark of the immune system, ensuring that T-cell responses are appropriately directed. Generally, TCRs on CD8+ cytotoxic T cells bind endogenous peptide antigens presented by class I MHC proteins, whereas TCRs on CD4+ helper T cells bind exogenous peptide antigens presented by class II MHC proteins. The basis for MHC class restriction is still widely discussed but is believed to emerge from the thymic development process together with the influence of a genetically encoded bias of TCRs towards MHC proteins[1–14].

Despite the hallmark of MHC class restriction, reports of CD4+ T cells that recognize antigens presented by class I instead of class II MHC proteins, and vice versa, date back nearly 40 years[15,16]. Such "class-mismatched" T cells and TCRs have since been described in the context of viruses, cancer, autoimmunity, and transplantation[17–27]. However, few studies have characterized the molecular properties of mismatched TCRs and the extent to which they follow other paradigms of TCR molecular recognition. Yin and colleagues described the structures of a single TCR bound to both class I and class II MHC

[1]Department of Chemistry & Biochemistry and the Harper Cancer Research Institute, University of Notre Dame, Notre Dame, IN, USA. [2]Department of Surgery, Cardinal Bernardin Cancer Center, Loyola University Chicago, Maywood, IL, USA. [3]Present address: Ragon Institute of MGH, MIT, and Harvard, Cambridge, MA, USA. [4]Present address: AbbVie, Inc. 1N Waukegan Rd, North Chicago, IL, USA. [5]Present address: Nature Technology Corporation, 4701 Innovation Drive 103, Lincoln, NE, USA. [6]Present address: Amgen Inc., One Amgen Center Drive, Thousand Oaks, CA, USA. [7]Present address: Harvard Medical School, Division of Gastroenterology, Boston Children's Hospital, Boston, MA, USA. [8]Present address: Department of Physical and Life Sciences, Nevada State College, Henderson, NV, USA. ✉e-mail: brian-baker@nd.edu

proteins[28], but this receptor was generated in an engineered mouse model and may not reflect naturally occurring biology[29].

One of the first naturally occurring mismatched TCRs to be identified, termed TIL1383I, was cloned from a tumor-infiltrating lymphocyte of a patient with metastatic melanoma[17]. The immunobiology of TIL1383I has been studied extensively. Although cloned from a $CD4^+$ T cell, TIL1383I recognizes the tyrosinase 368–376 tumor antigen naturally deamidated at position 3 ($Tyr_{370D}$; sequence YMDGTMSQV) presented by the class I MHC protein HLA-A*02:01 (referred to as HLA-A2). As with other instances of mismatched MHC recognition[27], TIL1383I is believed to have been selected on a class II protein (although class II ligands are unknown). TIL1383I gene transduction confers potent and specific $Tyr_{370D}$/HLA-A2 reactivity to both $CD4^+$ and $CD8^+$ T cells[30,31], consistent with an intrinsic independence for the CD8 coreceptor, although CD8 is still capable of enhancing TIL1383I signaling[32]. TIL1383I binds $Tyr_{370D}$/HLA-A2 with strong affinity and mediates anti-tumor immunity in mice and humans; indeed, its use in a clinical trial of gene-engineered T cells for the treatment of metastatic melanoma led to objective clinical responses in two patients in the first cohort, with no evidence of off-target reactivity[33–35].

Curious about the structural properties of TIL1383I, we aimed to uncover how traditional MHC restriction is overcome in a naturally occurring TCR. We find here that TIL1383I binds its class I ligand with a highly atypical docking geometry that places the TCR and its CDR loops in unusual positions over the MHC protein, attributable in part to an exceptionally long CDR3β loop. Having escaped filtering mechanisms that help establish MHC restriction, TIL1383I presents the case of a rare TCR that achieves an unorthodox but functional molecular pairing with a mismatched MHC. Our findings expand the range of known TCR-binding geometries that are compatible with high specificity and normal T-cell function, provide insight into the determinants of MHC restriction, and may help guide TCR selection and engineering for immunotherapy.

## Results

### TIL1383I binds Tyr370D/HLA-A2 with an atypical geometry

To understand how the TIL1383I TCR bound the class I MHC protein HLA-A2, we determined the X-ray crystal structure of the TIL1383I-$Tyr_{370D}$/HLA-A2 complex at 2.5 Å resolution (Supplementary Table 1 and Fig. 1). Upon examining the structure, it was apparent that the TCR bound $Tyr_{370D}$/HLA-A2 with an atypical geometry (Fig. 2a). In naturally occurring complexes of TCRs with class I MHC proteins, the α chain is usually positioned such that the germline-encoded CDR1α and CDR2α loops interact with the N-terminal half of the peptide and the long arm

of the MHC α2 helix, respectively[36]. CDR2β is almost invariably positioned over the α1 helix. This orientation reflects the "diagonal" TCR-binding mode and polarity most often observed in TCR-peptide/MHC structures. In the structure with TIL1383I, however, the TCR is rotated counterclockwise relative to this common positioning. Although traditional polarity is retained (Vα over the α2 helix; Vβ over the α1 helix), CDR1α is positioned near the center of the peptide and α2 helix, near the position normally occupied by CDR2α. CDR2α in turn is shifted to an unusual position over the short arm of the α2 helix. CDR1β and CDR2β are rotated away from the peptide and α1 helix, forming unusual contacts as described below.

Although the TIL1383I TCR is rotated relative to traditional structures, the hypervariable CDR3α and CDR3β loops still interact with the peptide. The architecture of CDR3β is particularly notable: the loop is positioned like a "lid" over the C-terminal portions of the peptide, forming numerous peptide contacts (Fig. 2a and Supplementary Fig. 1a). However, CDR3β also lies over the C-terminal half of the α1 helix, adopting the roles typically played by CDR1β and CDR2β and contacting both α helices. The position of CDR3β is in part a function of its exceptional length, which at 20 amino acids is approximately three standard deviations longer than the average CDR3β loop length for human HLA-A2 restricted TCRs, as well as class I and class II human MHC-restricted TCRs in general ($\bar{x} = 13.9 \pm 1.9$ amino acids for all tabulated HLA-A2 restricted TCRs, $14.1 \pm 1.9$ for human class I-restricted TCRs, and $14.5 \pm 1.6$ amino acids for human class II-restricted TCRs)[37]. Approximately 0.6% of tabulated human TCRs have CDR3β loop lengths of 20 or more amino acids. As CDR3β loop lengths approximate a normal distribution, this matches the prediction from the empirical rule that the frequency of loops of this length should be below 1%.

The diagonal rotation of a TCR over peptide/MHC is often described by the crossing angle, which quantifies the rotation of the variable domains relative to the peptide. The established means of measuring this angle uses a line drawn between the centroids of the variable domain disulfide bonds and a line fit to the peptide α carbons[38]. By this definition, TIL1383I binds with a nearly orthogonal angle of 81°, more than two standard deviations above the mean of $46 \pm 17°$ that describes TCRs binding to class I MHC proteins, and more than three standard deviations above the mean of $44 \pm 12°$ that describes the smaller set of TCRs bound to HLA-A2 (Fig. 2b). The TIL1383I crossing angle is also well above the average of $44 \pm 20°$ for TCR complexes with class II ligands. The crossing angle of TIL1383I is matched or exceeded only by unusually binding TCRs associated with autoimmunity or that bind with reverse polarity[39–43]. If we instead use

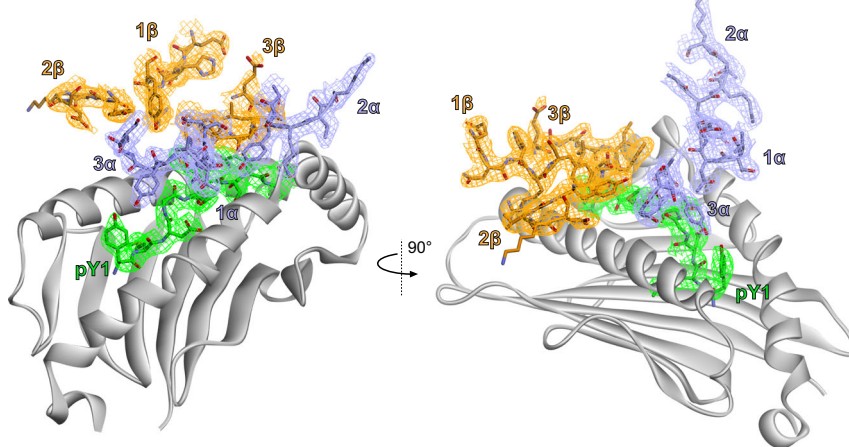

**Fig. 1 | Electron density for key regions of the TIL1383I complex.** Densities are from a $2F_o$-$F_c$ composite OMIT map contoured at 1σ, highlighting the TIL1383I CDR loops and peptide in the TCR-peptide/MHC structure. The color scheme used here is maintained through the following figures.

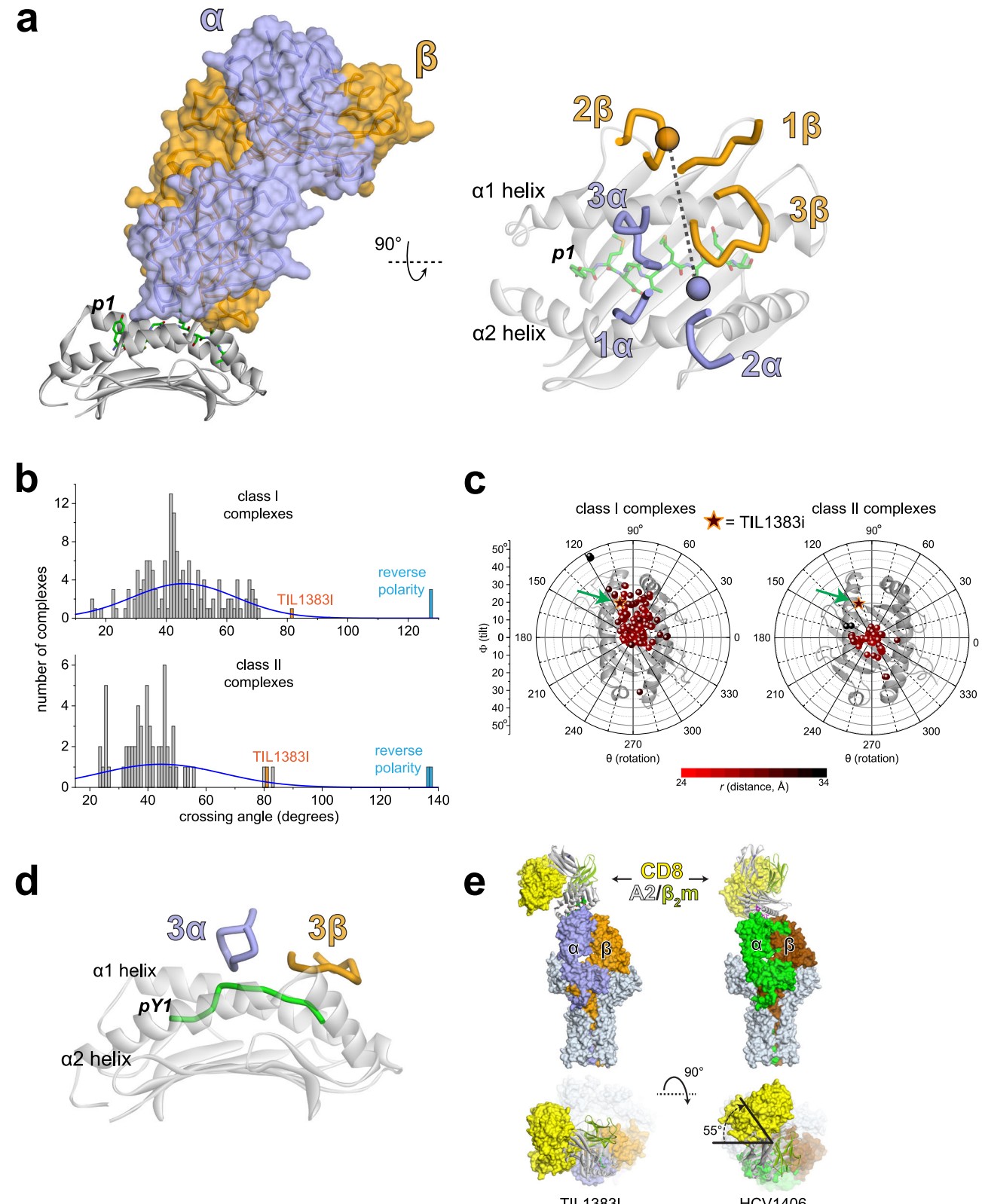

an angle measured using the centers of mass of the variable domains, as is commonly performed when visualizing TCR binding and shown in Fig. 2a, TIL1383I binds with an obtuse angle of 99°. By either definition, TIL1383I binds with a crossing angle far outside of what is traditionally seen in TCR-peptide/MHC complexes.

We recently described a polar coordinate method that provides additional criteria to describe TCR-binding geometries. Independent

of the crossing angle, this method gives the overall rotation ($\theta$), tilt ($\phi$), and distance ($r$) of the TCR variable domain center of mass (COM) relative to the MHC peptide-binding groove COM[44]. A key observation we made was a divergence in how TCRs recognize class I and class II MHC proteins, traceable to differences in how the two classes of proteins present peptides. TCRs that recognize class I MHC proteins bind in a way that allows the hypervariable CDR3 loops to access the

**Fig. 2 | Structural overview of the TIL1383I-Tyr$_{370D}$/HLA-A2 complex. a** TIL1383I binds Tyr$_{370D}$/HLA-A2 counterclockwise compared to traditional TCR-peptide/MHC complexes, placing the germline CDR loops in unusual positions. The left panel shows the overview of the TIL1383I complex. The right panel shows a view of the CDR loops over the peptide/HLA-A2 complex. Circles show the positions of the centers of mass of the TCR variable domains. **b** Histograms of crossing angles of TCRs bound to class I (top) or class II (bottom) peptide/MHC complexes. The value for TIL1383I, indicated in orange, is more than two standard deviations above the means for both classes. The values at the edges (>120°) are for recently described TCRs that bind with reverse polarity. **c** Quantitative geometrical analysis indicates that, despite binding atypically, TIL1383I still resembles a class I-restricted TCR in how it accesses the Tyr$_{370D}$ peptide. The position of TIL1383I is indicated by the star, also highlighted by the green arrow. The plots indicate the rotation ($\theta$), tilt ($\phi$), and distance ($r$) of the TCR variable domain COM relative to the MHC peptide-binding groove COM. **d** The class I-restricted character of TIL1383I revealed in panel **c** emerges from how the hypervariable CDR3 loops focus on the center and C-terminal bulge of the Tyr$_{370D}$ peptide. **e** Modeling the structure of the TIL1383I-Tyr$_{370D}$/HLA-A2 complex and the canonically binding HCV1406-NS3/HLA-A2 complex into the structure of the intact TCR/CD3 complex along with the structure of CD8 bound to HLA-A2 shows an approximate 55° difference in the placement of the CD8 coreceptor relative to the CD3 subunits. Data for panels **b** and **c** are provided as a Source Data file.

bulge found in the C-terminal halves of class I MHC-presented peptides. TCRs that recognize class II MHC proteins, on the other hand, bind in a way that allows their CDR3 loops to access the centers of the flat peptides presented by class II proteins. When analyzed via this method, its atypical crossing angle notwithstanding, TIL1383I binds in a more class I-like fashion, reflecting how the CDR3 loops access the bulge in the tyrosinase peptide (Fig. 2c, d).

Although TIL1383I recognition of Tyr$_{370D}$ is functionally independent of the CD8 coreceptor, CD8 is still capable of engaging the TIL1383I-Tyr$_{370D}$/HLA-A2 complex and enhancing signaling[32]. We have observed similar effects with other coreceptor-independent TCRs, including the HLA-A2-restricted TCR HCV1406, which recognizes the NS3 epitope from the Hepatitis C virus presented by HLA-A2[45,46]. Compared to TIL1383I, HCV1406 is a traditionally (or canonically) binding TCR, with a crossing angle of 30° (see ref. [47]). As shown by recent studies of TCRs that bind with reverse polarity, the unusual positioning of CD8 relative to the CD3 subunits can impact T-cell function[42]. To examine how TIL1383I binding would differentially place the CD8 coreceptor, we modeled both the TIL1383I and HCV1406 complexes into the recently determined structure of the intact TCR/CD3 complex. We then modeled in CD8 using the CD8/HLA-A2 structure[48,49]. This modeling indicated that the two TCRs result in a 55° difference in the placement of CD8 (Fig. 2e). As CD8 enhances TIL1383I signaling, this indicates that, provided traditional TCR polarity is maintained, normal CD8 function is compatible with a wide range of positioning relative to the TCR/CD3 complex.

## TIL1383I forms unusual contacts with HLA-A2

Within the TIL1383I-Tyr$_{370D}$/HLA-A2 interface, the majority of the contacts are between the TCR and the MHC protein. Interatomic contacts are tabulated in Supplementary Fig. 1a, many of which are unusual for TCR interfaces. For example, on the HLA-A2 α1 helix, the amino acids that typically interact with Arg65 are instead positioned over Arg75, down three turns of the helix and over 16 Å away from Arg65. Arg75 and nearby Gln72 are the foci of a complex electrostatic network involving Glu101 of CDR3β (Fig. 3a). Most unusually, a salt bridge is present between Arg34 of CDR1β and Glu19 at the end of one of the β-strands extending from the floor of the peptide-binding groove (Fig. 3a). This Vβ-MHC contact has not been observed in traditional TCR complexes with class I MHC proteins, although the neighboring Glu18 forms a hydrogen bond with the CDR3α loop of a TCR that binds the mouse class I MHC H-2D$^b$ with reverse polarity[43].

Contacts to the α2 helix are also shifted from where they are typically seen in TCR complexes with class I MHC proteins, with TCR contacts reaching from the center of the helix into the α2 helix short arm and its linker region (His151–Lys146). Polar interactions again dominate, particularly with Gln155, which interacts with all three CDR loops of the α chain (Fig. 3b).

As noted above, the hypervariable CDR3α and CDR3β loops of TIL1383I still interact with the bulge of the tyrosinase peptide despite the atypical TCR-binding geometry. The protruding methionine at peptide position 6 is accommodated in a cleft formed from the tips of the two hypervariable loops (Fig. 3c). Contacts between the TCR and

the peptide are focused on the peptide center and C-terminal half, with CDR3α contacting the sidechains of pMet6 (p referring to peptide) and pThr5, as well as the backbone of pGly4. CDR3β, in addition to interacting with HLA-A2, forms contacts from pMet6 through pGln8.

To test the importance of the length of CDR3β and our structural interpretations, we shortened CDR3β by eliminating Glu102β, Gly103β, and Gly104β, which lie towards the N-terminal edge of the loop as it enters the framework region. As expected from the structure, the deletion abrogated detectable TIL1383I binding as measured by surface plasmon resonance (SPR; Supplementary Fig. 2a). We also mutated Glu101β and Ile106β to alanine, which make substantial contacts to HLA-A2 and the peptide, respectively (Supplementary Fig. 1a). Again consistent with the structure, both CDR3β mutations substantially weakened TCR binding (Supplementary Fig. 2a). We also made alanine mutations for Asn100α and Tyr101α in CDR3α, which contact both HLA-A2 and the peptide (Supplementary Fig. 1a). These mutations also substantially weakened binding (Supplementary Fig. 2b), again consistent with our structural observations.

## TIL1383I does not rely on characteristic or shared interactions with HLA-A2

The growth in the number of published TCR-peptide/MHC structures has facilitated the identification of common or shared interactions between TCRs and MHC proteins[1–12,50]. In studying how TCRs recognize class I MHC proteins, we previously found that TCR complexes with HLA-A2 are distinguished by electrostatic interactions between the receptor and the polymorphic Arg65 on the HLA-A2 α1 helix[7]. These interactions arise as Arg65 is almost invariably buried upon binding, requiring a corresponding negative charge in the TCR[7]. Owing to the high prevalence of HLA-A2 in human populations, we found that TCR Vα genes are enriched in the negative charges needed to accommodate to Arg65. TIL1383I, however, avoids the charge on Arg65 altogether: no short- or long-range electrostatic interactions are formed with Arg65, and the residue's sidechain is mostly solvent-exposed (Fig. 4a). Other common interactions between TCRs and HLA-A2 are also absent. For example, Tyr52 in CDR2β is positioned above and does not interact with the HLA-A2 α1 helix, unlike most other TCR complexes with HLA-A2[7].

To validate this unusual structural observation, we mutated Arg65 in the HLA-A2 α1 helix to alanine. We tested the effect of the mutation on the binding of the TIL1383I TCR as well as the HCV1406 TCR, which as described above binds the NS3/HLA-A2 complex with a traditional binding orientation and forms multiple interactions with Arg65[47]. As determined by SPR, the mutation only moderately impacted binding of TIL1383I to Tyr$_{370D}$/HLA-A2 but substantially impaired binding of HCV1406 (Supplementary Fig. 3a), similar to what has been seen with other TCRs that bind HLA-A2 with traditional orientations[51,52].

Structures of different TCRs that share α- or β variable gene segments bound to the same MHC protein have revealed repeated interatomic contacts. TIL1383I shares *TRBV10-3* with the DMF4 and 38−10 TCRs, which recognize the MART-1 tumor antigen and a p53-derived neoantigen presented by HLA-A2, respectively. Structures are available for both of these complexes, which display traditional

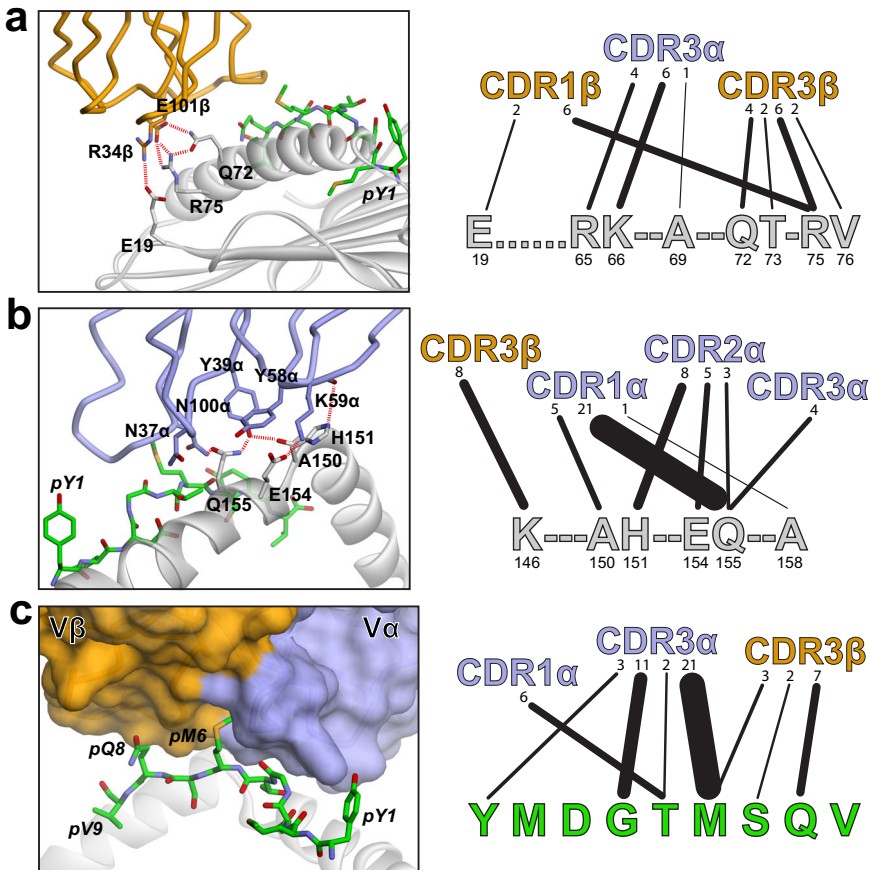

**Fig. 3 | Contacts between TIL1383I and the HLA-A2 α1 helix, α2 helix, and the Tyr₃₇₀ᴅ peptide. a** Gln72 and Arg75 on the HLA-A2 α1 helix participate in a complex network of electrostatic interactions involving Glu101 in the TIL1383I CDR3β loop, as shown in the left panel. An unusual salt bridge is also formed between Arg34 in CDR1β and Glu19 at the edge of the HLA-A2 β-sheet platform. The right image illustrates the contacts between the HLA-A2 α1 helix and the TCR. Linewidths are proportional to the number of contacts, enumerated above each line. **b** TCR contacts to the HLA-A2 α2 helix are shifted towards the helix short arm and the connecting linker region. Images are as in panel **a**, with the left emphasizing the reliance on electrostatic interactions and the right showing contacts made by the CDR loops to helix. **c** TCR contacts to the peptide are distributed from pTyr1 to pGln8, with pMet6 inserted into a deep pocket formed by CDR3α and CDR3β.

binding geometries[53,54]. Despite the shared *TRBV* gene, there are no Vβ-HLA-A2 contacts common to the TIL1383I, DMF4, and 38–10 complexes (Fig. 4b and Supplementary Fig. 1a, b). Although both the DMF4 and 38−10 TCR use Asp54 in CDR2β to form a salt bridge with Arg75 on the HLA-A2 α1 helix, TIL1383I offsets the Arg75 positive charge instead with Glu101 in CDR3β (Supplementary Fig. 1c). There are no structures of other TCRs bound to HLA-A2 that share the *TRAV4-1* of TIL1383I, although structures are available for *TRAV4-1* TCRs bound to HLA-B and HLA-C proteins, as well as a class II HLA-DQ2 protein[55–57]. Again, there are no conserved interactions among the structures. Overall then, the TIL1383I complex with Tyr₃₇₀ᴅ/HLA-A2 does not include any intermolecular interactions common to other TCR-peptide/MHC complexes with shared *TRAV* or *TRBV* genes.

To ask if HLA-A2 could be serving as a "molecular mimic" of a class II MHC protein, we performed a structural alignment between HLA-A2 and a representative class II protein to identify the amino acids in the class II α1 and β1 helices that equated to the most commonly contacted HLA-A2 amino acids in the TIL1383I structure (≥8 contacts; Arg75, Lys66, Gln155, His151, and Lys146) (Supplementary Fig. 4a). As the class II proteins expressed by the patient from which TIL1383I was cloned have not been reported, we compared these positions to known human class II HLA-DP, -DQ, and -DR genes[58]. The only point of identity is with Gln155 in HLA-A2, whose equivalent class II amino acid (position 70 in the β1 helix) is glutamine in 43% of HLA-DRB genes (Supplementary Fig. 4b). There is some charge conservation (e.g., position 65β

in class II, which equates to His151 in HLA-A2, is lysine in HLA-DQβ genes), although, when considered across the whole dataset, differences outweigh similarities. Moreover, although class I and class II MHC proteins are structurally similar, the conformations and positions of the class I α1 and class II β1 helices differ in and around the segments that join their short and long arms, with the class II helix extending further above the plane of the peptide-binding groove. As shown above, TIL1383I makes substantial contacts to this region of HLA-A2. Aligning a structure for HLA-DQ2.5[55] onto HLA-A2 in the TIL1383I-Tyr₃₇₀ᴅ/HLA-A2 complex showed that, without major structural rearrangements, considerable interatomic overlaps and clashes would occur with the TIL1383I CDR1α and CDR2α loops (Fig. 4c). As class II MHC proteins show structural variation in this region of the β1 helix[59], we repeated the comparison using a structure for HLA-DR2a. Although the structure of HLA-DR2a diverges from that of HLA-DQ2.5 in this region[60], we observed a similar degree of overlaps and clashes. Thus, HLA-A2 is not acting as a mimic of a class II protein, and binding of TIL1383I to a selecting or activating class II ligand thus requires a different TCR-binding geometry than seen with HLA-A2.

### High peptide specificity emerges from close packing in the TIL1383I-Tyr₃₇₀ᴅ/HLA-A2 interface

Studies in mice and humans using TIL1383I for T-cell therapy for melanoma have indicated high specificity with no off-target toxicity[33,34]. As conserved TCR-MHC interactions have been

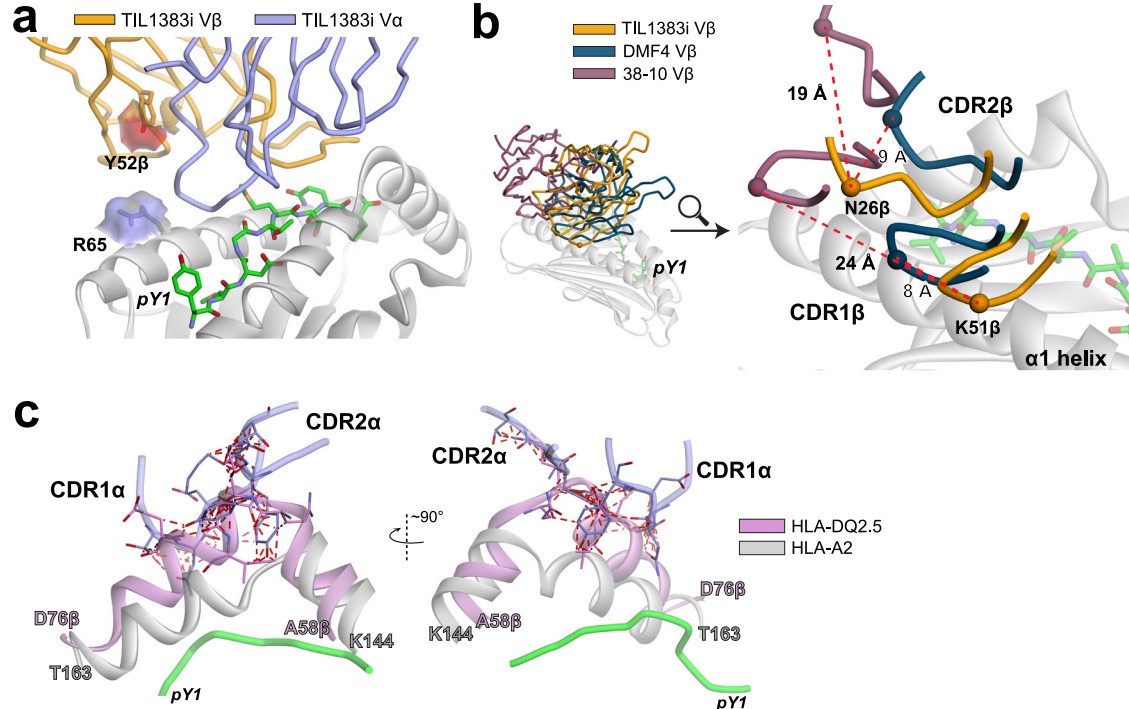

**Fig. 4 | TIL1383I avoids common or shared TCR-HLA-A2 contacts. a** TIL1383I does not interact with the charge of the Arg65 sidechain on the HLA-A2 α2 helix when bound to Tyr$_{370D}$/HLA-A2, nor does it use Tyr52 in CDR2β to contact HLA-A2. The solvent-exposed surface areas of the Arg65 and Tyr52β sidechains are shown, emphasizing their lack of participation in the interface. **b** There are no shared Vβ-HLA-A2 contacts between TIL1383I and two other TCRs that share *TRBV10-3* owing to the different placements of the CDR1β and CDR2β loops. **c** HLA-A2 is not serving as a molecular mimic of a class II protein, as without substantial rearrangements, the binding geometry of TIL1383I would result in significant interatomic steric clashes with the elevated β1 helix of a class II protein, shown by the red lines when the structure of HLA-DQ2.5 is aligned with HLA-A2 in the TIL1383I-Tyr$_{370D}$/HLA-A2 complex.

suggested to limit TCR cross-reactivity[4] and are lacking in the TIL1383I structure, we nonetheless aimed to quantitatively assess TCR specificity. We generated a positional scanning library of the tyrosinase peptide, varying each peptide position except the two primary anchors to all 20 natural amino acids. We first examined the library in functional experiments in which TIL1383I-expressing Jurkat cells co-expressing the CD8αα coreceptor were mixed with peptide-pulsed antigen-presenting T2 cells and IL-2 production assessed by ELISA. High specificity was seen for positions in 3–6, where most substitutions resulted in substantially lower cytokine production (Fig. 5a). There was very good agreement between the library data and prior alanine scanning results with TIL1383I recognition of Tyr$_{370D}$[30].

To examine specificity more quantitatively, we assessed this library biochemically. Using a UV-mediated peptide exchange protocol[61], we generated 140 soluble peptide/HLA-A2 complexes, each containing a member of the peptide library, with the native peptide represented seven times. We then developed a high throughput TCR-binding assay that permitted the rapid determination of $K_D$ values. To determine the concentrations of UV-exchanged samples, we took advantage of the previously described single chain TCR variant (scTv) S3-4 that binds peptide/HLA-A2 complexes in a peptide-independent manner[62]. Together with pre-determined activities of SPR sensor surfaces, this permitted measurements of the $K_D$ values for each member of the library (Supplementary Fig. 5). Experiments with the native Tyr$_{370D}$ peptide yielded values comparable to those determined in separate full titration experiments: for exchanged complexes, the $K_D$ was $18 \pm 2\,\mu M$ (average and SD of seven measurements), and for full titrations with separately generated complexes, the $K_D$ was $16 \pm 5\,\mu M$ (average and SD of eight measurements), indicating the binding assay is accurate and precise (although we anticipate less accuracy for very weak binders with affinities near or in the mM range).

Data for the TCR-binding assay with the library are shown as binding free energies in Fig. 5b. There was good concurrence with the functional experiments, as specificity was again very high for the region of the peptide ranging from position 3 to position 6. While many peptide variants were tolerated by TIL1383I, most weakened binding (average $\Delta\Delta G° = 2.1 \pm 1.4$ kcal/mol), and no variant substantially improved binding compared to the native peptide.

We also measured the thermal stabilities of the 140 UV-exchanged peptide/HLA-A2 complexes using differential scanning fluorimetry, in which the melting temperature of the complex serves as a proxy for peptide-binding affinity[63]. In general, the peptide substitutions had little impact on peptide binding, as most complexes possessed $T_m$ values at or near the 62 °C value that characterizes the stability of the complex with the native peptide (Fig. 5c and Supplementary Fig. 6). The results from the functional and binding screen are thus attributable to how the TCR engages the peptide/HLA-A2 complex, rather than secondary effects on peptide binding to HLA-A2.

Examining the TIL1383I-Tyr$_{370D}$/HLA-A2 structure in detail reveals how the receptor would be resistant to substitutions in positions 3–6. pGly4 is packed closely against Tyr101 of CDR3α, such that even a substitution to alanine would result in a clash without protein or peptide reorganization (Fig. 5d). The sidechain of pThr5 is similarly nestled between the peptide backbone and the HLA-A2 α2 helix. Not only does this preclude bulkier amino acids at p5, the pThr5 sidechain hydroxyl forms numerous contacts and a hydrogen bond with Asn37 of CDR1α. pMet6, for which the highest specificity was seen, protrudes into a tight cavity formed predominantly from Asn100, Gly102, and Gln105 of CDR3α and Phe107 and Pro108 of CDR3β (Fig. 5e). Bulkier sidechains again would not fit, and smaller sidechains would leave an unfavorable cavity or require distortions in the CDR loops. The other amino acid for which high specificity was seen, pAsp3, does not contact the TCR but is packed alongside Tyr159 and Tyr99 of HLA-A2, the

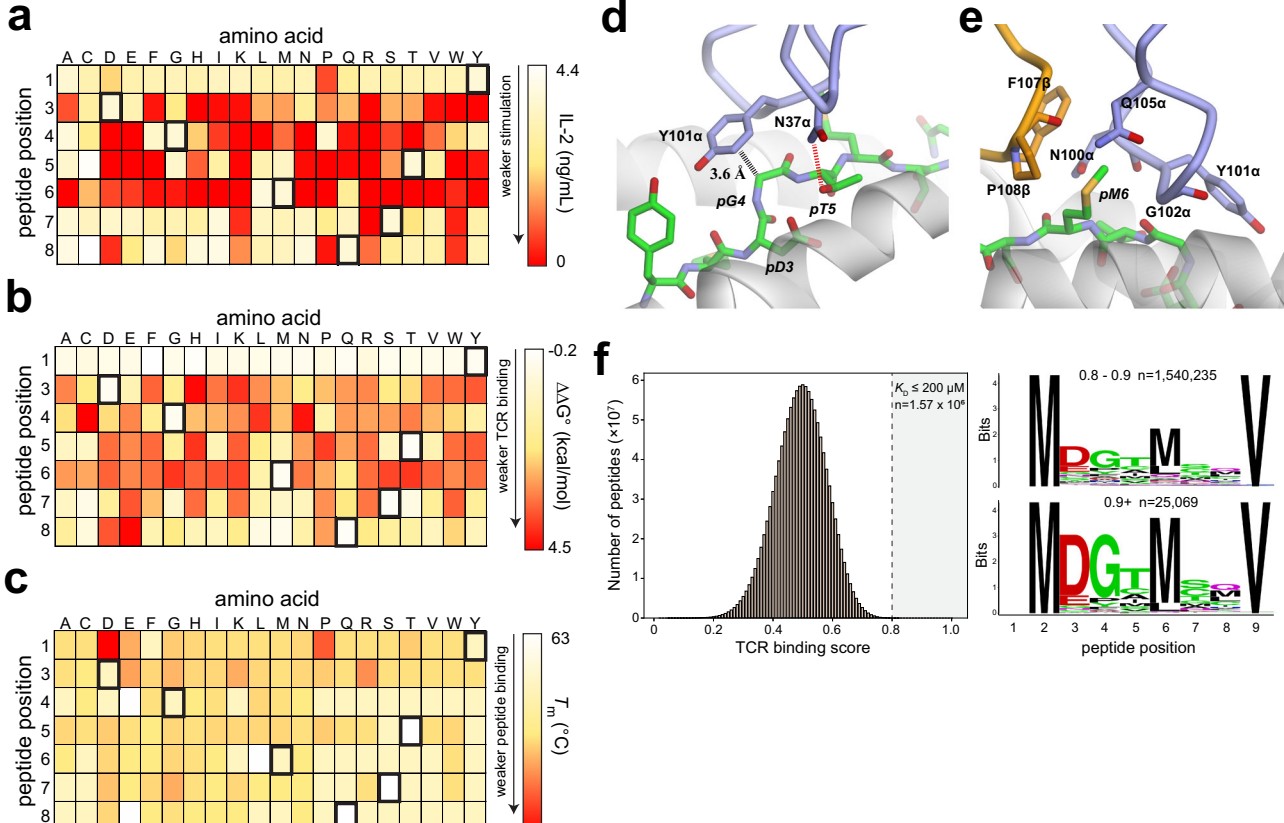

**Fig. 5 | Specificity within the TIL1383I·Tyr$_{370D}$/HLA-A2 interface. a** TIL1383I shows high specificity for the Tyr$_{370D}$ peptide, as shown by a functional analysis of a positional scanning library of the peptide. Cells of the heat map indicate the extent of IL-2 released when TCR-transduced Jurkat cells were mixed with peptide-pulsed HLA-A2$^+$ T2 cells. The color scale is on the right, with red indicating a loss of T-cell recognition. **b** Assessment of the positional scanning Tyr$_{370D}$ library by a novel TCR-binding assay with UV-exchanged peptide/HLA-A2 complexes. Cells of the heat map indicate the change in TCR-binding free energy relative to the wild-type peptide. The color scale is again on the right, with red indicating the largest reduction in binding free energy. There is good concurrence between the functional and binding analysis of the library. **c** Assessment of the UV-exchanged peptide/HLA-A2 library panel **b** using differential scanning fluorimetry to measure $T_m$ values. Cells of the heat map indicate the $T_m$. The color scale is again on the right, with red indicating the largest reduction in $T_m$. Most peptide variants have little impact on $T_m$ and thus peptide binding. **d, e** Structural determinants of high specificity for Asp3 through Met6 of the Tyr$_{370D}$ peptide in the interface with TIL1383I, showing how the interface would be intolerant to most substitutions at these peptide positions. **f** The binding data in panel **b** was used to generate a theoretical TCR-binding score for the 1.28 billion nonamers that retain methionine at position 2 and valine at position 9. The wild-type peptide has a score of 1.0, and the variant peptide scores have scores that are Gaussian-distributed around 0.5. There are 1.57 million peptides with a score of 0.8 or higher. The right panel shows sequence logos for the peptides in the top two deciles, along with the number of peptides. Data for panel **f** are provided as a Source Data file.

latter of which hydrogen bonds with the peptide backbone. Substitutions here likely have impacts on peptide conformation or dynamics, in turn impacting the close TCR-peptide fit.

We next used the library data to generate a specificity fingerprint for the TIL1383I TCR[64]. Binding free energies were used to construct a position weight matrix (PWM) which allowed any peptide sequence that retained the methionine and valine anchors at positions 2 and 9 to be scored relative to the native Tyr$_{370D}$ peptide. We used the PWM to score all theoretically possible 1.28 billion peptides. As seen with other systems[64], the scores were Gaussian-distributed near a score of 0.5. As the scores here were determined using binding affinities, we could directly relate scores to TCR-binding affinity (see "Methods"). This allowed us to estimate a $K_D$ of 5 mM for the peak score of 0.5, well outside the stimulatory range. The highest two deciles (scores ≥ 0.8) corresponded to a functionally relevant $K_D$ of 200 µM or stronger[65]. These deciles together contained 1,567,104 peptides, or ~0.1% of those theoretically possible (Fig. 5f). This number of peptides is notable, as TCRs have been estimated and in some cases shown to recognize at least 1 million different peptides[66–71].

To validate the specificity fingerprint, we selected 84 peptides across the range of scores. This included 72 peptides from the top 5% (scores ≥ 0.95, estimated $K_D$ values for TCR binding of 23–43 µM), 5 peptides at exactly 50% (scores of 0.5, estimated $K_D$ of 5 mM), and the 5 lowest-scoring peptides (scores < 0.2, estimated $K_D$ > 130 mM). The 72 peptides from the top 5% included the 24 highest-scoring peptides with two amino acid changes compared to the native Tyr$_{370D}$ peptide, the 24 highest-scoring peptides with three amino acid changes, and the 24 highest-scoring peptides with four amino acid changes. The peptides at the middle and bottom had 6–7 amino acid changes. The stimulatory capacity of these peptides was assessed in co-culture ELISA experiments measuring IL-2 production as in Fig. 5a. We saw excellent agreement between the scores and functional responses, with the 72 peptides in the top 5% producing an average of 90% of the amount of IL-2 as the native Tyr$_{370D}$ peptide (Fig. 6). All but one of the peptides in the middle and bottom ranges were inactive, with the sole active peptide only weakly stimulatory.

Having validated the specificity fingerprint, we searched the human proteome for peptides that matched those present in the top 20% (scores > 0.8), with the addition that we also allowed leucine to serve as the first primary anchor (p2). Only 146 peptides, or less than 0.01% of those in the top two deciles, were found. Thus, although as with most TCRs TIL1383I has the potential to cross-react with more than one million peptides, the number of actual compatible targets in

| | Peptide | # subs | IL-2 | Score | ΔG° | $K_D$ |
|---|---|---|---|---|---|---|
| WT | YMDGTMSQV | | 100 | 1.00 | -6.25 | 26 µM |
| Top 5% | FMDGTMSMV | 2 | 102 | 1.01 | -6.33 | 23 µM |
| | HMDGTMSLV | 2 | 95 | 1.00 | -6.26 | 26 µM |
| | NMDGTMSMV | 2 | 104 | 1.00 | -6.25 | 26 µM |
| | MMDGTMSMV | 2 | 103 | 1.00 | -6.24 | 27 µM |
| | FMDGTMTMV | 3 | 100 | 1.00 | -6.24 | 27 µM |
| | GMDGTMSMV | 2 | 103 | 1.00 | -6.24 | 27 µM |
| | LMDGTMSMV | 2 | 94 | 1.00 | -6.23 | 27 µM |
| | DMDGTMSMV | 2 | 81 | 1.00 | -6.23 | 27 µM |
| | QMDGTMSMV | 2 | 87 | 1.00 | -6.23 | 27 µM |
| | AMDGTMSMV | 2 | 93 | 1.00 | -6.22 | 27 µM |
| | EMDGTMSMV | 2 | 88 | 1.00 | -6.22 | 27 µM |
| | KMDGTMSMV | 2 | 100 | 0.99 | -6.22 | 28 µM |
| | IMDGTMSMV | 2 | 98 | 0.99 | -6.22 | 28 µM |
| | NMDGTMSLV | 2 | 102 | 0.99 | -6.21 | 28 µM |
| | HMDGTMTQV | 2 | 93 | 0.99 | -6.21 | 28 µM |
| | SMDGTMSMV | 2 | 98 | 0.99 | -6.21 | 28 µM |
| | MMDGTMSLV | 2 | 98 | 0.99 | -6.20 | 28 µM |
| | WMDGTMSMV | 2 | 85 | 0.99 | -6.20 | 29 µM |
| | FMDGTMTLV | 3 | 101 | 0.99 | -6.20 | 29 µM |
| | GMDGTMSLV | 2 | 91 | 0.99 | -6.20 | 29 µM |
| | VMDGTMSMV | 2 | 89 | 0.99 | -6.19 | 29 µM |
| | FMDGTMGQV | 2 | 90 | 0.99 | -6.19 | 29 µM |
| | LMDGTMSLV | 2 | 91 | 0.99 | -6.19 | 29 µM |
| | DMDGTMSLV | 2 | 86 | 0.99 | -6.19 | 29 µM |
| | QMDGTMSLV | 2 | 98 | 0.99 | -6.19 | 29 µM |
| | AMDGTMSLV | 2 | 97 | 0.99 | -6.18 | 29 µM |
| | HMDGTMGMV | 3 | 74 | 0.99 | -6.16 | 31 µM |
| | NMDGTMTMV | 3 | 96 | 0.99 | -6.16 | 31 µM |
| | FMDGTMAMV | 3 | 86 | 0.98 | -6.15 | 31 µM |
| | MMDGTMTMV | 3 | 93 | 0.98 | -6.15 | 31 µM |
| | FMDGTMGLV | 3 | 90 | 0.98 | -6.15 | 31 µM |
| | GMDGTMTMV | 3 | 88 | 0.98 | -6.15 | 31 µM |
| | HMDGTMMMV | 3 | 95 | 0.98 | -6.14 | 31 µM |
| | LMDGTMTMV | 3 | 89 | 0.98 | -6.14 | 32 µM |
| | DMDGTMTMV | 3 | 88 | 0.98 | -6.14 | 32 µM |
| | QMDGTMTMV | 3 | 92 | 0.98 | -6.14 | 32 µM |
| | AMDGTMTMV | 3 | 102 | 0.98 | -6.13 | 32 µM |
| | EMDGTMTMV | 3 | 89 | 0.98 | -6.13 | 32 µM |
| | KMDGTMTMV | 3 | 96 | 0.98 | -6.13 | 32 µM |
| | IMDGTMTMV | 3 | 92 | 0.98 | -6.13 | 32 µM |
| | NMDGTMTLV | 3 | 95 | 0.98 | -6.12 | 33 µM |

| | Peptide | # subs | IL-2 | Score | ΔG° | $K_D$ |
|---|---|---|---|---|---|---|
| WT | YMDGTMSQV | | 100 | 1.00 | -6.25 | 26 µM |
| Top 5% | HMDGAMSMV | 3 | 87 | 0.98 | -6.12 | 33 µM |
| | SMDGTMTMV | 3 | 90 | 0.98 | -6.12 | 33 µM |
| | MMDGTMTLV | 3 | 97 | 0.98 | -6.11 | 33 µM |
| | NMDGTMGMV | 3 | 70 | 0.98 | -6.11 | 33 µM |
| | FMDGIMSMV | 3 | 90 | 0.98 | -6.11 | 33 µM |
| | WMDGTMTMV | 3 | 94 | 0.98 | -6.11 | 33 µM |
| | GMDGTMTLV | 3 | 94 | 0.98 | -6.11 | 33 µM |
| | HMDGAMTMV | 4 | 97 | 0.96 | -6.03 | 38 µM |
| | FMDGIMTMV | 4 | 79 | 0.96 | -6.02 | 39 µM |
| | FMDGAMGMV | 4 | 56 | 0.96 | -6.01 | 39 µM |
| | FMEGTMTMV | 4 | 93 | 0.96 | -6.00 | 40 µM |
| | FMDGAMMMV | 4 | 88 | 0.96 | -5.99 | 40 µM |
| | HMDGAMTLV | 4 | 92 | 0.96 | -5.99 | 41 µM |
| | HMDGIMTMV | 4 | 31 | 0.96 | -5.99 | 41 µM |
| | FMDGSMTMV | 4 | 88 | 0.96 | -5.99 | 41 µM |
| | HMDGAMGMV | 4 | 66 | 0.96 | -5.98 | 41 µM |
| | NMDGAMTMV | 4 | 83 | 0.96 | -5.98 | 41 µM |
| | FMDGAMAMV | 4 | 94 | 0.96 | -5.98 | 42 µM |
| | MMDGAMTMV | 4 | 88 | 0.96 | -5.97 | 42 µM |
| | FMDGAMGLV | 4 | 83 | 0.96 | -5.97 | 42 µM |
| | HMEGTMTMV | 4 | 67 | 0.96 | -5.97 | 42 µM |
| | GMDGAMTMV | 4 | 99 | 0.95 | -5.97 | 42 µM |
| | HMDGMMTMV | 4 | 96 | 0.95 | -5.97 | 42 µM |
| | FMEGTMTLV | 4 | 95 | 0.95 | -5.96 | 43 µM |
| | LMDGAMTMV | 4 | 95 | 0.95 | -5.96 | 43 µM |
| | DMDGAMTMV | 4 | 19 | 0.95 | -5.96 | 43 µM |
| | HMDGSMTMV | 4 | 98 | 0.95 | -5.96 | 43 µM |
| | FMDGAMMLV | 4 | 98 | 0.95 | -5.95 | 43 µM |
| | AMDGAMTMV | 4 | 92 | 0.95 | -5.95 | 43 µM |
| | FMEGTMGMV | 4 | 0 | 0.95 | -5.95 | 43 µM |
| | EMDGAMTMV | 4 | 103 | 0.95 | -5.95 | 43 µM |
| 50% | MMWDRYQQV | 6 | 0 | 0.50 | -3.13 | 5 mM |
| | AMHRYENQV | 6 | 0 | 0.50 | -3.13 | 5 mM |
| | IMDQFTPNV | 6 | 0 | 0.50 | -3.13 | 5 mM |
| | KMLVYQIAV | 7 | 22 | 0.50 | -3.13 | 5 mM |
| | LMFGYNWVV | 7 | 0 | 0.50 | -3.13 | 5 mM |
| Bottom 20% | SMHNPTWEV | 7 | 0 | 0.19 | -1.19 | 133 mM |
| | TMHNPTEEV | 7 | 0 | 0.19 | -1.19 | 133 mM |
| | WMHNPTWEV | 7 | 0 | 0.19 | -1.18 | 136 mM |
| | VMHNPTWEV | 7 | 0 | 0.19 | -1.18 | 137 mM |
| | RMHNPTREV | 7 | 0 | 0.19 | -1.16 | 140 mM |
| neg CTRL | LLFGYPVYV | 9 | 0 | 0.00 | | |

**Fig. 6 | Experimental validation of the TIL1383I specificity analysis.** In total, 72 peptides from the top 5% (scores ≥ 0.95) in the specificity analysis in Fig. 5f were selected for evaluation in co-culture experiments measuring IL-2 production. Peptides selected included the 24 highest-scoring peptides with two amino acid changes relative to the native $Tyr_{370D}$ peptide, the 24 highest-scoring peptides with three amino acid changes, and the 24 highest-scoring peptides with four amino acid changes. Five peptides at 50% (scores = 0.5, 6–7 amino acid changes) were also selected, as were the five lowest-scoring peptides (bottom 20%, scores = 0.19, 7 amino acid changes). The sequence of each peptide is listed. Amino acid changes relative to the $Tyr_{370D}$ peptide are in bold red font. The number of amino acid substitutions is given in the "# subs" column. IL-2 produced normalized to the amount produced by the WT peptide is colored red (maximum) to white (zero), along with the percent. The score from the fingerprint analysis is indicated by the "score" column. The estimated ΔG° (in kcal/mol) and $K_D$ for TCR binding, computed from the score as described in the Methods, are also indicated. "neg CTRL" indicates a negative control peptide unrelated to $Tyr_{370D}$.

humans is very low. Although this analysis is limiting in that it assumes each peptide position contributes independently to TIL1383I binding, does not assess other peptide lengths, and ignores class II MHC ligands, it nonetheless highlights the high specificity of the TIL1383I TCR and is consistent with clinical outcomes showing no off-target toxicity.

# Discussion

Reports of T cells with αβ TCRs that violate the norms of MHC class restriction date back at least 40 years[15,16]. Class-mismatched T cells or TCRs have since been described in the context of viruses, cancer, autoimmunity, and transplantation[17–27]. Although seemingly rare, they

are clearly a component of the natural repertoire. Developing T cells include a pool of receptors that are broadly cross-reactive and bind both class I and class II MHC, but these are believed to be largely eliminated during lineage commitment and thymic education[29]. The existence of successful class mismatched T cells in the periphery—those that demonstrate high specificity and are signaling competent—could thus reflect a specific subset of cells that have escaped thymic development mechanisms, or they could be rare instances of pedestrian, class-restricted T cells that express a TCR nonetheless cross-reactive with a mismatched MHC ligand.

In the case of TIL1383I, a substantial clue to its origin is found in its exceptionally long CDR3β loop, more than three standard deviations above the average. In generating MHC-restricted TCRs, thymic education has been shown to bias TCR repertoires away from those with long hypervariable loops[72–74]. Developing T cells expressing TCRs with longer loops are more likely to undergo death by neglect, indicating that longer loops are less compatible with MHC recognition[72]. When they do bind, long CDR3 loops have also been suggested to yield TCR-binding modes less reliant on commonly-engaged MHC features[50]. These are our precise observations: TIL1383I binds $Tyr_{370D}$/HLA-A2 unusually, utilizing an outlier binding geometry that avoids the most frequently seen contacts TCRs form with HLA-A2, including those believed to be influenced by evolution. In addition, the complex with TIL1383I shares no common contacts with those of other TCRs with known structures that utilize the same germline genes. Altogether, this leads us to conclude that TIL1383I represents an unusual TCR that escaped a key filtering process of thymic education, leaving it with properties distinct from the majority of mature TCRs, including the ability to bind with high specificity to both class I and class II peptide/MHC complexes. By extension, our findings are in strong support of a major role for thymic education in establishing traditional MHC restriction.

By escaping the filtering processes of thymic education and binding atypically, we believe that TIL1383I recognition of $Tyr_{370D}$/HLA-A2 is opportunistic: the unique structural and chemical makeup of the TIL1383I binding site is serendipitously structurally and physically compatible with the $Tyr_{370D}$/HLA-A2 composite surface. However, although binding is unusual, it is still compatible with TCR signaling. Such functionally opportunistic molecular recognition has been hypothesized to explain rare TCRs that bind alternative ligands independent of MHC, as well as those comprising the repertoire in MHC-deficient mice[75]. A related concept has been proposed to explain unusual binding with autoreactive TCRs, in which opportunistic binding leads to pathogenesis in the presence of high amounts of self antigen[40,76,77]. In the original melanoma patient from which TIL1383I was cloned, upregulation of the tyrosinase gene may have served a similar purpose, allowing an otherwise quiescent T cell to expand, drive tumor infiltration, and contribute to tumor rejection.

Although they suggest limits to its influence, our results do not rule out a genetically encoded predisposition of TCRs for MHC proteins in stabilizing the complex between TIL1383I and $Tyr_{370D}$/HLA-A2. The outcome of TCR-MHC coevolution is expected to be structurally malleable to accommodate the extensive diversity in TCR-peptide/MHC interfaces[3,7,9,11,12], involving features such as long-range charge complementarity, "slippery" hydrophobic character, and shape complementarity, leading to what we have referred to as an evolved biomolecular compatibility[7]. Further work is needed to assess the roles these play in facilitating TIL1383I binding and how features such as CDR3 loop length and composition tune their contributions.

Although peptide specificity is usually considered a hallmark of T-cell immunology, mature TCRs have been estimated to be compatible with at least one million different peptide/MHC complexes[66–68], an estimate which has been supported experimentally[69–71]. This degree of cross-reactivity is needed given the limited size of the T-cell repertoire relative to the vast universe of potential antigens. Our specificity analysis of TIL1383I also found it to be functionally compatible with 1–2

million different peptides, albeit only a small number of these are present within the human proteome. Thus, the specificity and cross-reactivity possessed by most TCRs is not a consequence of traditional TCR-binding geometries. Rather, we believe it to be a consequence of how TCRs pack against the relatively flat peptide/MHC surface, irrespective of binding orientation and how the CDR loops are used. This biophysical and structural plasticity, coupled with moderate affinities that bias against the formation of very strong interatomic interactions, permits TCR cross-recognition with ligands sharing structural and physicochemical similarities.

There is significant discussion about the relationship between TCR-binding geometries and T-cell signaling and function. For example, TCRs that bind with reverse polarity have been shown to signal aberrantly, due in part to altered positioning of the CD8 coreceptor, which in turn impacts recruitment and utilization of the Lck kinase[42]. It is thus notable that despite its highly atypical binding geometry over $Tyr_{370D}$/HLA-A2, TIL1383I signals normally. Moreover, despite placing CD8 more than 50° away from its position with a canonically docked TCR, TIL1383I can utilize CD8 to enhance signaling. The results here thus significantly expand the known range of fully functional TCR-binding geometries, which should be accounted for when considering TCR signaling mechanisms.

We hypothesize that other class-mismatched TCRs that escape the biologic mechanisms that drive traditional MHC restriction will possess attributes similar to TIL1383I. Given their coreceptor independence and potential for high specificity and high potency, such TCRs may be ideal lead candidates for TCR-based immunotherapy or suggest opportunities and novel approaches for TCR engineering.

## Methods

### Recombinant protein production and purification

Recombinant TIL1383I, HCV1406, peptide/HLA-A2, and S3-4 for analysis and crystallography were produced and purified as previously described[62,78]. Briefly, TCR α and β chains, the HLA-A2 heavy chain, β2-microglobulin (β2m), and the S3-4 scTv were expressed in *E. coli* as inclusion bodies and dissolved in 8 M urea and 6 M guanidinium-HCl. Proteins were generated by refolding from solubilized inclusion bodies. For TIL1383I, HCV1406, and S3-4 refolding, inclusion bodies for the TCR α and β chains at a 1:1 ratio or the S3-4 scTv were diluted into TCR refolding buffer (50 mM Tris-HCl, 2.5 M urea, 2 mM EDTA, 9.6 mM cysteamine, 5.5 mM cystamine, and 0.2 mM PMSF, pH 8) and incubated at 4 °C overnight. The refolding reaction was dialyzed against 10 mM Tris-HCl (pH 8.3) at 4 °C for 36 h. Refolded protein was purified using a DEAE anion exchange column, followed by S200 and S75 size exclusion chromatography. Purified protein was maintained in 10 mM HEPES, 150 mM NaCl, pH 7.4. For peptide/HLA-A2 refolding, the HLA-A2 heavy chain and β2m at a 1:1 ratio were diluted into MHC refolding buffer (100 mM Tris-HCl, 400 mM L-arginine, 2 mM EDTA, 3 mM cysteamine, 3.7 mM cystamine, 0.2 mM PMSF, pH 8) in the presence of tenfold molar excess peptide. Twelve hours post-incubation, the solution was dialyzed against 10 mM Tris-HCl pH 8.15 for 36–48 h at room temperature. The refolded complex was then purified as above. The TIL1383I and HCV1406 constructs included an engineered disulfide bond across the α and β chain constant domains for increased stability[79]. TCR mutations were generated using PCR mutagenesis and confirmed by sequencing; sequencing results are available as described in the Data Availability section. Oligonucleotides used for mutagenesis are provided as Supplementary Table 2. The Arg65 to alanine mutation in HLA-A2 was previously described[80]. Expression vectors for TCRs and HLA-A2 variants are available upon request.

### Peptides

Individual peptides were purchased from AAPTEC or Genscript at >80% purity. The 140-member positional scanning peptide library was purchased from AAPTEC. The library had positions two and nine fixed

at methionine and valine, respectively, with all other positions substituted with the 20 naturally occurring amino acids. Peptides were stored at −80 °C in DMSO until use.

## Crystallization, structure solution, and analysis

Crystals of the complex between TIL1383I and Tyr$_{370D}$/HLA-A2 were grown in 16% PEG 3350, 2% tacsimate, buffered in 100 mM Tris (pH 8.5) at 25 °C. Crystallization was performed using hanging-drop vapor diffusion. Protein concentration was 6 mg/mL with both TCR and peptide/HLA-A2 at an equimolar ratio. For cryoprotection, crystals were transferred into 20% glycerol with 80% mother liquor for 30 s and immediately frozen in liquid nitrogen. X-ray diffraction data was collected at the 23ID-D GM/CA beamline at the Advanced Photon Source at Argonne National Laboratories. The structure was solved by molecular replacement using Phaser in Phenix[81], using PDB 3QDM from which the HLA-A2 heavy chain and TCR variable domains were used as search models[54]. Rigid-body refinement followed by NCS torsion-angle restraints, translation/libration/screw refinement, and multiple steps of restrained refinement were performed using Phenix Refine[82]. Anisotropic and bulk solvent corrections were considered throughout the refinement process. Evaluation of models and fitting to maps were performed using COOT[83]. Molprobity was used to evaluate the structure during and after refinement[84]. As seen before[47], several segments of the α3 domain of HLA-A2 were poorly resolved and were excluded from the final model. The composite OMIT map was calculated with CNS[85] as implemented in Discovery Studio 2021. Structural analysis was performed with PyMOL and Discovery Studio. Geometrical binding parameters were calculated as previously described[44], using tools available at the TCR3d database (https://tcr3d.ibbr.umd.edu)[86]. The data for TCR crossing angles and other geometrical parameters were obtained from the TCR3d database, incorporating data available through August 2022 for mouse and human TCRs. Geometrical data for non-redundant structures involving engineered receptors were removed. Solvent-accessible surface areas were calculated with a 1.4 Å probe radius. Interatomic contacts were calculated with Contpro web server using a 4 Å cutoff[87].

## CDR3β loop length analysis and structural alignments and comparisons

CDR3β loop sequences were obtained from the VDJdb online database (https://vdjdb.cdr3.net)[37], using data current as of August 2022. Sequences were obtained by (1) filtering on β chains, humans, class I MHC, and HLA*02:01, which yielded 15,274 sequences, (2) filtering on β chains, humans, and class I MHC, which yielded 71,799 sequences, and (3) filtering on β chains, humans, and class II MHC, which yielded 2112 sequences, upon which we assumed control. CDR3β loop lengths are inclusive of the N-terminal cysteine and C-terminal phenylalanine. To compare HLA-A2 with class II HLA structures and evaluate steric clashes with TIL1383I, HLA-A2 was structurally aligned onto HLA-DQ2.5 from PDB ID 4OZI[55] and HLA-DR2a from PDB ID 1ZGL[60] using the align structures tool in Discovery Studio 2021. HLA class II amino acids at structurally equivalent sites were tabulated using the IMGT database (https://www.imgt.org)[58]. Modeling of the TCR/CD3/HLA-A2/CD8αα complexes was performed by aligning the TCRs from the structures of the TIL1383I and HCV1406 complexes (PDB ID 5JZI)[47] on the TCR in the structure of TCR/CD3 complex (PDB ID 6JXR)[48], and then superimposing HLA-A2 from the structure of the CD8αα/HLA-A2 complex (PDB ID 1AKJ)[49] onto HLA-A2 in the TIL1383I and HCV1406 complexes. The α3 domain from the CD8αα/HLA-A2 complex was used to aid in visualization.

## Cell lines, media reagents, and ELISA

Jurkat 76 and T2 cell lines were maintained in RPMI-1640 media supplemented with 10% FBS, 100 units/mL penicillin, and 100 μg/mL

streptomycin. The generation of CD8$^+$/CD34$^-$ Jurkat 76 cells have been previously described[32]. Cells were transduced using the Neon Transfection System (Thermo Fisher) with the pCMV(CAT)T7-SB100 plasmid and the pSBbi-Neo Sleeping Beauty vector bearing the full length TIL1383I TCR α and β chains separated by the P2A self-cleaving peptide as previously described[78]. Stable transformants were positively selected by culturing cells in complete media containing 1.2 mg/mL of G418. Prior to co-culture experiments, TCR-expressing cells were monitored via flow cytometry for CD8 and TCR cell surface expression by staining cells with anti-human CD8 Alexa Fluor-conjugated antibody and anti-human CD3 APC/Cy7-conjugated antibody (BioLegend #344726 at 2.5 mg/mL and BioLegend #300426 at 10 mg/mL, respectively). A figure exemplifying gating strategies is provided as Supplementary Fig. 7. Screening of the Tyr$_{370D}$ positional scanning library and the individual peptides was performed by pulsing $1 \times 10^5$ T2 cells with a 100 μM or 1 μM of each peptide, respectively, for 2 h at 37 °C, then adding an equal number TCR-expressing CD8$^+$ Jurkat cells and incubating co-cultures at 37 °C for 18–20 h as previously described[78]. Phorbol myristate acetate at 25–50 ng/mL was added to TCR-positive cells prior to co-culture to increase sensitivity to stimulation. IL-2 release was measured by ELISA and replicated twice. Expression vectors and cell lines are available upon request.

## Production of UV-exchanged peptide/HLA-A2 library

For the production of the UV-exchanged peptide/HLA-A2 library, recombinant HLA-A2 bound to the UV-cleavable peptide KILGFVFJ*V, where J* is 3-amino-3-(2-nitro)phenyl-propionic acid[61], was produced as described above. Wild-type human β$_2$m was substituted with a mutant containing a histidine-rich tag in the β$_2$m FG loop (sequence NHVTLSQ beginning at position 84 mutated to HHHTLHH). Once purified, 160 μL of photocleavable peptide/HLA-A2 at approximately 10 μM in HBS buffer (10 mM HEPES, 150 mM NaCl, pH 7.4) was dispensed into the wells of a 96-well plate (two plates were used to cover the 140-member library). Plates were placed on ice and exposed to UV light using a Spectrolinker UV oven equipped with 365 nm UV lamps in 5-minute intervals for a total of 45–60 min, with approximately 5 min between intervals during which the plates were gently shaken on ice. After UV exposure, library peptide was added to yield a final concentration of approximately 1 mM peptide along with DMSO to a final concentration of 20% to promote peptide solubility. Exchange reactions were subsequently incubated at room temperature for two hours. Following the incubation, 50 μL Ni$^{2+}$-charged magnetic bead slurry was added to each well, and the exchanges incubated overnight at 4 °C. Exchange reactions were transferred to a 96-well plate with an internal 0.2-micron filter and centrifuged at 800×$g$ for 3 min to separate protein-bound beads from the solution. Beads were washed in the plate with 200 μL cold HBS and the plate centrifuged again to remove the wash. The wash step was repeated 3–4 times. Intact peptide/HLA-A2 was removed from the magnetic beads by adding 100 μL HBS supplemented with 3 mM EDTA and 0.005% surfactant P20 and incubating overnight at 4 °C. The protein solution was separated from the beads by centrifugation at 800 × $g$ for 3 min.

## Surface plasmon resonance

Surface plasmon resonance experiments were performed on a Biacore T200 instrument. For individual titrations with TIL1383I, HCV1406, and S3-4, experiments were performed as previously described[78]. Briefly, TCR was amine coupled to a sensor surface and Tyr$_{370D}$/HLA-A2 or NS3/HLA-A2 injected at increasing concentrations at a flow rate of 5 μL/ min at 25 °C until steady state was reached (250 s). Coupling densities were between 1000 and 5000 resonance units (RU). The responses were corrected by subtracting the response of a blank flow cell. Data were processed with BiaEvaluation 4.1 and fit using a 1:1 binding model in MATLAB 2017a or OriginPro 2019. Titrations with HLA-A2/TCR mutants were globally fit alongside titrations with wild-type molecules

using a shared $RU_{max}$ to increase accuracy as previously described[88]. Experiments were performed at 25 °C in HBS-EP (10 mM HEPES, 150 mM NaCl, 3 mM EDTA, 0.005% surfactant p20, pH 7.4). Full titrations with TIL1383I and the S3-4 scTv were repeated at least three times ($n = 8$ for TIL1383I, $n = 3$ for S3-4). Depending on available sample, some titrations included duplicate injections which were treated as a single dataset and fit together. The average $K_D$ values for TIL1383I and S3-4 were nearly identical to those previously determined[34,62].

For analysis of the UV-exchanged library, TIL1383I was again immobilized on a sensor surface using standard amine coupling to approximately 5000 resonance units. The scTv S3-4, which binds HLA-A2 independently of peptide[62], was similarly immobilized on an adjacent flow cell. Samples were injected serially over the flow cells for 250 s at a flow rate of 5 µL/min at 25 °C and corrected for the response over a blank flow cell. Experiments were performed in HBS-EP at 25 °C. Data were processed with BiaEvaluation 4.1. Concentrations of exchanged peptide/HLA-A2 were calculated using the $K_D$ and $RU_{max}$ determined from a separate analysis of the S3-4 surface with $Tyr_{370D}$/HLA-A2 as diagrammed in Supplementary Fig. 5. The average concentration of UV-exchanged samples was $14 \pm 4$ µM. The concentration of the UV-exchanged sample was used along with the separately determined $RU_{max}$ of the TIL1383I surface to determine the $K_D$ as diagrammed in Supplementary Fig. 5. For presentation in Fig. 5 and the subsequent specificity analysis, $K_D$ values were converted to binding $\Delta G°$ values, using the standard equation $\Delta G° = RT\ln K_D$, where $R$ is the gas constant and $T$ is the temperature in Kelvin. $\Delta\Delta G°$ values relative to the wild-type peptide were calculated using the average value for the wild-type peptide. Exchanges with $Tyr_{370D}$ were measured 7 times. Errors in the UV-exchanged values were propagated from the errors in the $K_D$ from the S3-4 titrations and $RU_{max}$ from the TIL1383I titration with WT $Tyr_{370D}$ and converted to errors in $\Delta G°$ using standard error propagation methods. Errors in $\Delta G°$ ranged from 0.03 to 0.04 kcal/mol.

**Position weight matrix and evaluation of theoretical peptides**

The PWM was calculated from the library $\Delta G°$ values in a manner analogous to that described by Karapetyan et al.[64]. $\Delta G°$ values were converted to a percent response of the range using $100 \times (\Delta G° - min)/(max - min)$, where $min$ is the minimum (strongest) $\Delta G°$ value observed and $max$ is the weakest. The PWM thus consists of the percent response for each member of the library. A normalization factor was determined from the percent response of the sum over each amino acid in the WT peptide. Scores for any theoretical peptide matching the motif XMXXXXXXV, where X is any of the 20 standard amino acids, were then determined by finding the percent response for the individual amino acids at each peptide position in the PWM, summing these for the theoretical peptide, and dividing by the normalization factor. The resulting scores for the 1.28 billion peptides were binned with a size of 0.01. Sequence logos for peptides scoring in the 8th and 9th deciles were constructed using WebLogo[89]. All human sequences in the NCBI non-redundant protein database[90] were searched for peptides scoring at or above 0.8, with the addition that leucine was also permitted at peptide position 2. The estimated affinity for peptide scores was computed from $K_D = \exp(s(\Delta G°_{WT}/RT))$, where $s$ is the score and $\Delta G°_{WT}$ is the TCR binding free energy for the WT $Tyr_{370D}$ peptide.

**Differential scanning fluorimetry**

DSF on the UV-exchanged library was performed as previously described using an Applied Biosystems StepOnePlus RT-PCR instrument using excitation and emission wavelengths of 587 nm and 607 nm, respectively[63]. SYPRO orange at approximately tenfold greater than the protein concentration for a total volume of 20 µL. The temperature increment was from 25 to 95 °C at a scan rate of 1 °C per minute. Data were analyzed by taking the first derivative of the melting curve, and the $T_m$ approximated as the temperature at the maximum of the derivative curve.

**Reporting summary**

Further information on research design is available in the Nature Portfolio Reporting Summary linked to this article.

## Data availability

The structural data generated in this study have been deposited in the PDB database under accession code 7RK7. Other data (sequencing, oligonucleotides, and SPR data) are available at the Zenodo data repository at https://zenodo.org/record/7259584 or are available in the Source data file provided with this paper. Source data are provided with this paper.

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

## Acknowledgements

Supported by grant R35GM118166 from NIGMS, NIH. J.A.A., G.L.J.K., and A.M.R. were supported by fellowships from the Indiana CTSI (grant UL1TR002529). GM/CA at the Advanced Photon Source has been funded by the National Cancer Institute (ACB-12002) and the National Institute of General Medical Sciences (AGM-12006, P30GM138396). This research used resources of the Advanced Photon Source, a U.S. Department of Energy (DOE) Office of Science User Facility operated for the DOE Office of Science by Argonne National Laboratory under Contract No. DE-AC02-06CH11357.

## Author contributions

Structural biology was performed by N.K.S., G.L.J.K., L.M.L., A.G.A., and L.I.W. Binding measurements were performed by J.R.D., A.M.R., and A.K.C. The UV-exchange-based binding assay was developed and performed by J.R.D. Quantitative analysis of the library data was performed by J.R.D. and G.L.J.K. Measurements of thermal stability were performed by J.R.D. T-cell functional experiments were performed by J.A.A., G.I.G., and S.G.F. L.M.H. assisted in recombinant protein production and analysis. The TIL1383I TCR and Jurkat cells were provided by M.I.N. B.M.B. obtained funding and oversaw the project. The manuscript was written and edited by all authors.

## Competing interests

The authors declare no competing interests.
