## [Peer Review File · Nature Communications]

A class-mismatched TCR bypasses the logic of MHC restriction via an unorthodox but fully functional binding geometryREVIEWER COMMENTS

Reviewer #1 tumour antigen presentation (Remarks to the Author):

This manuscript explores binding of a mismatched CD4 TCR (TIL13831) to a class I MHC HLA-A2 presenting a tyrosinase tumor antigen. While binding is highly specific, it has an atypical geometry with unusual contacts. The authors suggest the recognition is opportunistic in nature.

The manuscript is well written with background information included so that it is understandable to a board readership. The findings are well supported by the data, which includes appropriate controls. The clinical relevance of the recognition is also presented.

Minor:

Please define what the “p” stands for in TCR-pMHC at first presentation (page 5, line 101).

Page 13, lines 274-275: please define “p” in pThr5 and p5

The first two sentences of the discussion are a repeat from the intro.

Reviewer #2 TCR/MHC structure (Remarks to the Author):

This manuscript by Singh et al. describes the structural basis by which a TCR is class-mismatched - that is, although the TCR derives from a CD4+ T cell, and thus should recognize a peptide presented by an MHC class II molecule, this TCR, TIL13831, also recognizes a peptide (Tyrosinase; Tyr[370D]) presented by the MHC class I molecule HLA-A2. The authors present a high-resolution crystal structure and do some rather elegant functional and binding experiments with a positional scanning peptide library.

From their structure of the peptide-MHC/TCR complex, the authors conclude that the especially long CDR3beta loop confers binding and specificity of the TCR to the Tyr[370D]/HLA-A2 complex. However, they do not show this experimentally. Demonstrating that mutations in the CDR3beta loop and/or versions of TIL13831 bearing shortened CDR3beta loops result in reduced T cell stimulation and/or peptide-MHC complex binding would go a long way to support this conclusion.

The authors also present a very nice position weight matrix (PWM) analysis in which they predict that roughly 0.1% of the 1.28 billion possible peptides with similar anchor residues as Tyr[370D] would bind to TIL13831. This is in line with numerous reports in the literature indicating a similar number of peptides are recognized by any given TCR. Here, again, the authors pass up some low hanging experimental fruit. To back up their computational analysis, they could choose just a handful of the 1.5M+ peptides that they predict to bind their TCR and show that they do so experimentally. Showing that a couple from the lowest two deciles do not bind would be nice as well.

Reviewer #3 MHC restriction/T cells (Remarks to the Author):

This manuscript from Brian Baker's laboratory by Singh et al presents a structural study of how an unconventional MHC-mismatched TCR from a human CD4 T cell binds to its MHC-I/peptide ligand. Unlike the MHC-mismatched TCR studies in this manuscript, conventional MHC-restricted TCRs engage MHC/peptide ligands with a typical binding geometry that involves germline-encoded TCR-CDR1 and TCR-CDR2 sequences interacting with MHC sequences while non-germline TCR-CDR3 sequences interact primarily with the MHC-bound peptide. This manuscript now documents for an MHC-mismatched TCR (i.e. TIL1383I) that it binds to its MHC class I/peptide ligand with a very unusual geometry independently of CD4/CD8 coreceptors. In this mismatched TCR, the unusually long CDR3b sequence binds to MHC class I sequences as well as peptide sequences in accounting for this TCRs specificity for both MHC class I and peptide. The manuscript is well-written and the authors describe the unique binding structure clearly. I think the novel geometry of interaction between the mismatched TCR and its MHC class I/peptide ligand will be of significant interest to the scientific community.

Nevertheless, I am disappointed that the authors did not also consider the potential implications of their structural analysis for our understanding of MHC restriction. There exist two competing models of MHC-restriction: the germline model and the thymic selection model. The germline model postulates that TCR V-regions have evolved to encode TCR that only recognize MHC/peptide ligands and that germline-encoded TCR-CDR1 and TCR-CDR2 sequences promote MHC specificity by interacting directly with MHC sequences. In contrast the thymic selection model postulates that TCR sequences recombine to recognize random ligand specificities and that MHC restriction results from the fact that only MHC-restricted TCR are positively selected in the thymus based mainly on TCR-CDR3 sequences (see reference 74 of the manuscript). Interestingly, the germline model requires TCR-CDR3 sequences to be short, whereas the thymic selection model allows TCR-CDR3 sequences to be excessively long. In their current manuscript the authors simply accept all of the presumptions of the germline model and explain the novel binding geometry of their unique TCR as simply 'opportunistic', i.e. a one-off, without any further thought or discussion of how it is that their TCR was allowed to have an excessively long CDR3b region. The fact that they attribute their TCRs MHC class I/peptide binding specificity to excessively long CDR3b sequences would seem to give them reason to consider their TCR as a challenge to the germline model, rather than to simply consider it to be opportunistic. I think readers' interest in this manuscript would be enhanced by a discussion of the potential implications of this novel structure to different models of MHC restriction.

Response to reviewers

We thank the reviewers and editors for their comments on our manuscript. We appreciate that each of the three reviewers viewed our manuscript favorably, with requests for minor revisions but also new data to support our conclusions. In revising the manuscript, we have completed all requested revisions. This includes data for a series of new mutations that support our interpretations of the structural data, as well as data for a range of peptides that support our bioinformatic evaluation of specificity. We have also substantially revised the introduction and conclusion to address how the thymic education process shapes MHC restriction. Along with these requested revisions, we have made small changes to enhance readability and fix typos. Lastly, we appreciate the extended time allowed for revisions as completing the experiments requested by Reviewer 2 required hiring and training new staff. We hope our revised manuscript is reviewed favorably, and again thank the reviewers and editorial staff for their efforts.

Reviewer 1

“This manuscript explores binding of a mismatched CD4 TCR (TIL13831) to a class I MHC HLA-A2 presenting a tyrosinase tumor antigen. While binding is highly specific, it has an atypical geometry with unusual contacts. The authors suggest the recognition is opportunistic in nature.

The manuscript is well written with background information included so that it is understandable to a board readership. The findings are well supported by the data, which includes appropriate controls. The clinical relevance of the recognition is also presented.”

We thank the reviewer for their favorable view of our manuscript, including the potential clinical relevance.

“Please define what the “p” stands for in TCR-pMHC at first presentation (page 5, line 101).”

and

“Page 13, lines 274-275: please define “p” in pThr5 and p5”

In revising the manuscript, we have addressed these and other abbreviations.

“The first two sentences of the discussion are a repeat from the intro.”

This is correct, and we have respectfully left them unchanged, as the paper is complex, and we wished to start the discussion with a quick summary. We have however, made substantial changes in the discussion to address the results more directly, including removing various superfluous statements. We thank the reviewer for considering this approach.

Reviewer 2

“This manuscript by Singh et al. describes the structural basis by which a TCR is class-mismatched - that is, although the TCR derives from a CD4+ T cell, and thus should recognize a peptide presented by an MHC class II molecule, this TCR, TIL13831, also recognizes a peptide (Tyrosinase; Tyr[370D]) presented by the MHC class I molecule HLA-A2. The authors present a high-resolution crystal structure and do some rather elegant functional and binding experiments with a positional scanning peptide library.”

We thank the reviewer for their favorable summary of our manuscript.

“From their structure of the peptide-MHC/TCR complex, the authors conclude that the especially long CDR3beta loop confers binding and specificity of the TCR to the Tyr[370D]/HLA-A2 complex. However, they do not show this experimentally. Demonstrating that mutations in the CDR3beta loop and/or versions of TIL13831 bearing shortened CDR3beta loops result in reduced T cell stimulation and/or peptide-MHC complex binding would go a long way to support this conclusion.”

We agree that structural interpretations are always best supported with confirming experiments. As the reviewer requested, we mutated the CDR3 β (and CDR3 α) loops. We shortened the CDR3 β loop by three amino acids. As expected from the structure, this eliminated detectable binding. We also made individual mutations in both CDR3 loops, choosing sites which made substantial contacts to pMHC. Consistent with the structure, TCR binding affinity was substantially reduced for each individual mutation.

This new data is described in the Results on p. 9 and shown in a new supplementary figure (new Fig. S3).

“The authors also present a very nice position weight matrix (PWM) analysis in which they predict that roughly 0.1% of the 1.28 billion possible peptides with similar anchor residues as Tyr[370D] would bind to TIL13831. This is in line with numerous reports in the literature indicating a similar number of peptides are recognized by any given TCR. Here, again, the authors pass up some low hanging experimental fruit. To back up their computational analysis, they could choose just a handful of the 1.5M+ peptides that they predict to bind their TCR and show that they do so experimentally. Showing that a couple from the lowest two deciles do not bind would be nice as well.”

We appreciate this suggestion also and agree that it is a relatively simple set of experiments that would add substantial value to the paper. As requested, we selected 84 peptides to assay. 72 of these were in the top 5%, including the top 24 peptides with 1, 2, and 3 amino acid substitutions. In co-culture experiments with 13831 expressing Jurkats, the top 5% peptides almost all performed as expected, yielding in an average of 90% of IL-2 production compared to the WT peptide. We also selected 8 peptides from the middle and bottom deciles. These also perform as expected, with only one of the middle range peptides weakly stimulatory. We feel this is a strong validation of the specificity analysis and appreciate the value the reviewer's suggestion adds to the paper.

This new data is described in the Results on pp. 14-15 and shown in a new supplementary figure (new Fig. S8).

Reviewer 3

“This manuscript from Brian Baker's laboratory by Singh et al presents a structural study of how an unconventional MHC-mismatched TCR from a human CD4 T cell binds to its MHC-I/peptide ligand. Unlike the MHC-mismatched TCR studies in this manuscript, conventional MHC-restricted TCRs engage MHC/peptide ligands with a typical binding geometry that involves germline-encoded TCR-CDR1 and TCR-CDR2 sequences interacting with MHC sequences while non-germline TCR-CDR3 sequences interact primarily with the MHC-bound peptide. This manuscript now documents for an MHC-mismatched TCR

(i.e. TIL1383I) that it binds to its MHC class I/peptide ligand with a very unusual geometry independently of CD4/CD8 coreceptors. In this mismatched TCR, the unusually long CDR3b sequence binds to MHC class I sequences as well as peptide sequences in accounting for this TCRs specificity for both MHC class I and peptide. The manuscript is well-written and the authors describe the unique binding structure clearly. I think the novel geometry of interaction between the mismatched TCR and its MHC class I/peptide ligand will be of significant interest to the scientific community.”

We thank the reviewer for this favorable view.

“Nevertheless, I am disappointed that the authors did not also consider the potential implications of their structural analysis for our understanding of MHC restriction. There exist two competing models of MHC-restriction: the germline model and the thymic selection model. The germline model postulates that TCR V-regions have evolved to encode TCR that only recognize MHC/peptide ligands and that germline-encoded TCR-CDR1 and TCR-CDR2 sequences promote MHC specificity by interacting directly with MHC sequences. In contrast the thymic selection model postulates that TCR sequences recombine to recognize random ligand specificities and that MHC restriction results from the fact that only MHC-restricted TCR are positively selected in the thymus based mainly on TCR-CDR3 sequences (see reference 74 of the manuscript).”

We appreciate the summary of the two competing models. As described below and by many authors, these models are not mutually exclusive. At the same time though, we have published *significantly* about the shortcomings of the germline model as it is most commonly articulated, as discussed below and in the revised manuscript.

“Interestingly, the germline model requires TCR-CDR3 sequences to be short, whereas the thymic selection model allows TCR-CDR3 sequences to be excessively long. In their current manuscript the authors simply accept all of the presumptions of the germline model and explain the novel binding geometry of their unique TCR as simply ‘opportunistic’, i.e. a one-off, without any further thought or discussion of how it is that their TCR was allowed to have an excessively long CDR3b region. The fact that they attribute their TCRs MHC class I/peptide binding specificity to excessively long CDR3b sequences would seem to give them reason to consider their TCR as a challenge to the germline model, rather than to simply consider it to be opportunistic. I think readers’ interest in this manuscript would be enhanced by a discussion of the potential implications of this novel structure to different models of MHC restriction.”

We respectfully disagree with the first half of this statement (“...the authors simply accept all the presumptions of the germline model...”) – as noted above, we interpret the “germline model” as it is most commonly articulated to imply that genetics has instructed TCR germline regions to bind MHC proteins, just as the reviewer states. *We have disagreed with this interpretation and published arguments against it.* We have shown in prior work that although evolution has influenced TCR germline genes to be *compatible* with MHC proteins, and although this may contribute to a *bias* for TCRs towards MHCs, it is not necessarily driving and there are numerous ways a bias can be overcome (see ref. 7 in the revised manuscript). We have also published arguments against the common strict interpretation of the “roles” of CDR1/2/3 loops, as CDR1/2 loops quite often interact strongly with the peptide and CDR3 loops quite often interact strongly with the MHC (see ref. 55 in the revised manuscript). This fully leaves open the opportunity for TCRs to bind other things (including non-MHC proteins) as well as break MHC restriction.

The second statement of the reviewer however is correct: we did not address thymic education and how this TCR might have emerged. In revising the manuscript, we now explicitly addressed this exact point directly as described below.

“The fact that they attribute their TCRs MHC class I/peptide binding specificity to excessively long CDR3b sequences would seem to give them reason to consider their TCR as a challenge to the germline model, rather than to simply consider it to be opportunistic. I think readers’ interest in this manuscript would be enhanced by a discussion of the potential implications of this novel structure to different models of MHC restriction.”

In revising the manuscript, we have directly addressed how the long CDR3b loop contributes to mismatched binding as well the relationship to thymic selection. We agree with the reviewer that data indicating that long loops are biased against and result in death by neglect argue for a role of thymic education in establishing MHC selection and agree that our data supports this interpretation.

In response we have fully rewritten the Discussion. For example, on p. 16 we write:

Altogether, this leads us to conclude that TIL1383I represents an unusual TCR that escaped a key filtering process of thymic education, leaving it with properties distinct from the majority of mature TCRs, including the ability to bind with high specificity to both class I and class II peptide/MHC complexes. By extension, our findings are in strong support of a major role for thymic education in establishing traditional MHC restriction.

In the Introduction on pp. 4-5 we write:

Notably, thymic education has been predicted to bias TCRs towards traditional binding geometries and help establish MHC restriction by selecting for TCRs with short hypervariable loops (38-41). We thus conclude that TIL1383I represents a rare TCR that escaped a key biological filtering mechanism that contributes to traditional MHC restriction yet is serendipitously capable of binding a class I peptide/MHC ligand in a functional and specific manner.

Notably, thymic education has been predicted to bias TCRs towards traditional binding geometries and help establish MHC restriction by selecting for TCRs with short hypervariable loops (38-41). We thus conclude that TIL1383I represents a rare TCR that escaped a key biological filtering mechanism that contributes to traditional MHC restriction yet is serendipitously capable of binding a class I peptide/MHC ligand in a functional and specific manner.

We note that although we do not accept the germline model as commonly articulated, we cannot rule out the influence of TCR-MHC co-evolution on influencing binding (by, for example, positioning charges in regions that allow long range compensation, as we have published). However, we interpret co-evolution *from our own perspective* and temper the discussion based on the points made above. This is found on p. 17 of the Discussion where we write:

The outcome of TCR-MHC coevolution is expected to be structurally malleable to accommodate the extensive diversity in TCR-pMHC interfaces involving features such as

long-range charge complementarity, “slippery” hydrophobic character, and shape complementarity, leading to what we have referred to as an evolved biomolecular compatibility (7). Further work is needed to assess the roles these play in facilitating TIL1383I binding and how features such as CDR3 loop length and composition tune their contributions.

We hope this clarifies our thinking (including the much more restrictive interpretation of “germline bias”) and that our conclusions about a major role for thymic education in establishing MHC restriction are consistent with the reviewer’s requests.

REVIEWERS' COMMENTS

Reviewer #1 (Remarks to the Author):

The authors have adequately addressed my concerns and I approve the manuscript for publication.

Reviewer #2 (Remarks to the Author):

I am satisfied with the revisions made by the authors in response to my comments. All of my concerns have been addressed and, I believe, the manuscript more complete and much improved.

Reviewer #3 (Remarks to the Author):

The authors have now altered the manuscript to include a consideration of the germline and thymic selection models of MHC restriction. While I think these two models of MHC restriction are not as easily reconciled as the authors suggest, the revisions to the manuscript have definitely increased its interest for the scientific community. I strongly support its acceptance for publication.

Response to reviewers

The reviewers did not request any additional changes. We thank the reviewers and the editorial team for their efforts on our manuscript.